# Roles of ADP-Ribosylation during Infection Establishment by *Trypanosomatidae* Parasites

**DOI:** 10.3390/pathogens12050708

**Published:** 2023-05-12

**Authors:** Joshua Dowling, Craig L. Doig

**Affiliations:** School of Science & Technology, Nottingham Trent University, Nottingham NG11 8NS, UK; joshua.dowling@ntu.ac.uk

**Keywords:** ADP-ribosylation, PARP, PARG, *Trypanosoma*, *Leishmania*

## Abstract

ADP-ribosylation is a reversible post-translational protein modification, which is evolutionarily conserved in prokaryotic and eukaryotic organisms. It governs critical cellular functions, including, but not limited to cellular proliferation, differentiation, RNA translation, and genomic repair. The addition of one or multiple ADP-ribose moieties can be catalysed by poly(ADP-ribose) polymerase (PARP) enzymes, while in eukaryotic organisms, ADP-ribosylation can be reversed through the action of specific enzymes capable of ADP-ribose signalling regulation. In several lower eukaryotic organisms, including *Trypanosomatidae* parasites, ADP-ribosylation is thought to be important for infection establishment. *Trypanosomatidae* encompasses several human disease-causing pathogens, including *Trypanosoma cruzi*, *T. brucei*, and the *Leishmania* genus. These parasites are the etiological agents of Chagas disease, African trypanosomiasis (sleeping sickness), and leishmaniasis, respectively. Currently, licenced medications for these infections are outdated and often result in harmful side effects, and can be inaccessible to those carrying infections, due to them being classified as neglected tropical diseases (NTDs), meaning that many infected individuals will belong to already marginalised communities in countries already facing socioeconomic challenges. Consequently, funding to develop novel therapeutics for these infections is overlooked. As such, understanding the molecular mechanisms of infection, and how ADP-ribosylation facilitates infection establishment by these organisms may allow the identification of potential molecular interventions that would disrupt infection. In contrast to the complex ADP-ribosylation pathways in eukaryotes, the process of *Trypanosomatidae* is more linear, with the parasites only expressing one PARP enzyme, compared to the, at least, 17 genes that encode human PARP enzymes. If this simplified pathway can be understood and exploited, it may reveal new avenues for combatting *Trypanosomatidae* infection. This review will focus on the current state of knowledge on the importance of ADP-ribosylation in *Trypanosomatidae* during infection establishment in human hosts, and the potential therapeutic options that disrupting ADP-ribosylation may offer to combat *Trypanosomatidae*.

## 1. ADP-Ribosylation in Infection

ADP-ribosylation is a fundamental post-translational protein modification in which single or several ADP-ribose units are covalently attached to proteins. The modification is commonly catalysed by members of the poly(ADP-ribose) polymerase (PARP) enzyme family. PARP enzymes attach ADP-ribose moieties to the aspartate, glutamate, lysine, arginine, cysteine, threonine, or serine residues, resulting in the creation of branched and linear polymers [1]. In addition to the actions of the PARP enzymes, ADP-ribosylation can also occur via the action of mono(ADP-ribosyl)transferases, which catalyse the attachment of ADP-ribose to arginine side chains via the activity of an essential and highly conserved R-S-EXE motif [2]. This motif is localised within a specialised loop used in target recognition for mono(ADP-ribosyl) transferases as well as for several other ADP-ribosyltransferases (ARTs), including human PARP-1.

### 1.1. ADP-Ribosylation in Viral Pathogens

PARP enzymes have long-documented actions in infection progression and protection against pathogens in humans [3]. The role of PARP-mediated protection, particularly against viral infection, has seen extensive study. Several human PARPs, including PARP 1, 5, 7, 9, 10, 12, and 13 are known antagonists of both DNA and RNA viruses [4,5,6]. Viral replication is disrupted by several PARP enzymes, predominantly targeting dysregulation in viral genomic translation and transcription, leading to inhibition or prevention of the completion of the viral life cycle. PARP13, in particular, demonstrates potent antiviral activity against many DNA and RNA viruses, including alphaviruses, influenza, filoviruses (including Ebola and Marburg viruses), herpes virus, HIV-1, coxsackie virus B3, hepatitis B, and Japanese encephalitis virus [7,8,9,10]. PARP13 utilizes multiple mechanisms in viral inhibition, binding viral RNA through its four zinc-finger motifs, to allow the inhibition of transcription and translation of the viral genome which disrupts the viral life cycle. PARP13 is, then, able to degrade the 5′ end of HIV-1 RNA via the recruitment of several degrading host factors, including poly(A)-specific ribonuclease (PARN) and the KHNYN endonuclease [11].

Viral replication is targeted by the host expressing PARP enzymes to defend against infection. For example, PARP1 and PARP5 serve as antagonists to critical binding sites used by viral proteins for genomic replication by Kaposi sarcoma-associated herpesvirus (KSHV) and Epstein–Barr virus (EBV), respectively [12]. PARP1 is also able to prevent transcriptional elongation of HIV-1 RNA through the binding of PARP1 to a TAR binding site, which is responsible for the binding of RNA elongation factor p-TEFb. PARP1 also binds with TAR more efficiently than p-TEFb, demonstrating its ability to inhibit viral replication through epigenetic modification [13]. Genome translation is essential for viral replication and a target for inhibition by PARPs. PARP13 has been shown to decrease the production of Nef, a protein that is present in HIV-1 and is critical for successful viral replication. PARP13 is also able to degrade viral RNA via exosome activation [14]. This is further evidence of the broad antiviral mechanisms exhibited by PARP13. Other PARPs demonstrate antiviral activity through the targeted degradation of essential viral proteins. PARP9 can form a protein-degrading complex with a ubiquitin ligase capable of *Picronoviridae* protein degradation [15]. PARP10 is capable of transfer to nuclei during avian influenza (AIV) viral infection to degrade the protein AIV NS1, which is important for AIV replication [16]. PARP12 can degrade Zika virus (ZIKV) proteins NS1 and NS3 via mon(ADP-ribosyl)ation, catalysed by PARP12 [17].

### 1.2. ADP-Ribosylation in Bacterial Pathogens

Studies examining the roles of PARPs during bacterial pathogen infection are not as extensive in comparison to those on PARPs in viral infections. Only a small number of bacterial species possess functional PARylation systems. Although the majority contain domains capable of PAR binding, in addition to PAR-degrading enzymes [18]. The majority of antibacterial studies on PARP enzymes have primarily focused on the use of PARP1. Studies have been performed on several notable human bacterial pathogens, including *Helicobacter*, *Salmonella*, *Escherichia coli*, *Pseudomonas aeruginosa*, *Streptococcus pneumonia*, *Streptococcus pyogenes*, and *Chlamydophila*. Experiments utilising these bacteria surmise a similar consensus, whereby shifts in PARP activity increase the difficulty of mounting an effective response in preventing damage from bacterial infection [19,20,21,22,23]. In particular, the depletion of PARP activity depresses the sufficient inflammatory response to bacterial infection. This is likely due to PARP1 regulation of NF-kB-mediated signalling and the activation of macrophage responses [24,25,26].

Several species of bacteria have also been found to possess PARG enzymes, including *Thermomonospora curvata* and *Herpetosiphon aurantiacus* [27]. In humans, PARG enzymes are a mechanism through which PAR can be removed from the cell via catabolism of poly(ADP-ribose), through hydrolysis of the ribose-ribose bonds. This prevents damage caused by excessive PAR accumulation in the cytoplasm [28]. PAR accumulation can lead to a PARP-mediated cell death pathway known as parthanatos, through which excessive PAR can lead to apoptosis via several mechanisms, including depletion of NAD and the PAR-mediated activation of an apoptosis-inducing factor (AIF) [29]. PAR can bind to AIF, which is followed by AIF translocation to the nucleus, resulting in extensive DNA fragmentation and chromatin damage [30]. PARG is the primary means through which excessive PAR is removed in human cells. Other human PAR hydrolases do exist, necessitated by PARG’s inability to remove the most proximal ADP-ribose moieties [31], including ARH3, which is present during the removal of PAR from the mitochondria [32]. The network of PAR enzymes present in bacteria is much simpler in comparison. Human PARG enzymes act both as endo-glycohydrolases and exo-glcohydrolases, which leads to the production of free PAR moieties and mono-ADP ribose moieties, respectively, via the hydrolysis of ADP-ribose chains [28]. However, bacterial PARG enzymes are thought to be more limited in their roles, in that they are only capable of acting as exo-glcohydrolases due to the presence of a ribose cap located close to the C-terminus, which prevents PARG from binding efficiently to the internal binding sites necessary for endo-glycohydrolase activity [33]. Nevertheless, bacterial PARG enzymes are important to prevent PAR-mediated apoptosis, and as such, PARG inhibitors may present an interesting therapeutic option in the treatment of bacterial infections by limiting the pathogen’s ability to remove harmful PAR accumulation within the bacterial cytoplasm.

There is much evidence to suggest the extensive roles of human PARP enzymes in immune protection during bacterial and viral infection. However, key questions remain for both. PARP activity is seemingly broader across enzymes in terms of antiviral activity (10 out of 17 human PARP enzymes have identified antiviral activity), potentially as a result of primarily cytoplasmic and nuclear localisation, which allows the PARP enzymes to interrupt viral replication cycles at several distinct stages [34]. Therefore, it seems that the expression of multiple cytoplasmic PARP enzymes, developed alongside the evolution of vertebrates, are seemingly as equally as important as the nuclear-localised PARPs in maintaining cellular health through the maintenance of essential processes and antimicrobial activity. Given the broad antiviral properties of the most studied PARPs, there is considerable potential to further explore the actions of the remaining PARPs to better understand their antiviral properties.

The use of broad-spectrum PARP agonists may present an attractive avenue to explore novel antiviral therapies. ADP-ribosyltransferase (ART) activity of PARP enzymes has been demonstrated in works examining bacterial infection [35,36]. Further investigation is required to fully understand the range of ART activity in inflammatory responses and infection. Interestingly, inhibition of PARP activity in animal models has displayed therapeutic benefits. Moreover, the acute septic shock has been resolved as a result of PARP modulation, likely due to a reduction in tissue damage, which usually results from enhanced PARP-mediated ART activity [35]. This poses an interesting approach to treating bacterial infection by finding the correct balance of PARP activity versus PARP inhibition. However, excessive inhibition of PARP-catalysed ART activity would lead to inefficient DNA repair during bacterial infection and restrict the positive impacts of the PARP activity. However, PARP inhibition does yield beneficial therapeutic effects in defined circumstances. Combinatorial therapy with appropriate antibiotics used in conjunction with a lowered dose of PARP inhibitors could theoretically allow the successful compromise of PARP activity. This would allow a limitation upon PARP-driven DNA damage, whilst retaining the benefits that PARP activity yields during the infection response. Nevertheless, this process would require significant clinical testing and optimisation for different bacterial species. Given the current dearth of effective treatment options for Trypanosomes, all options must be worthy of exploration.

## 2. Trypanosomatidae

*Trypanosomatidae* is an order of singular flagellate kinetoplastid parasites, the most relevant to human health being *Trypanosoma* and *Leishmania*. The parasites *T. cruzi*, *T. brucei*, and *Leishmania* are all responsible for infections categorised by the World Health Organisation (WHO) as neglected tropical diseases (NTDs) [37]. As such, these infections are responsible for profound impacts as they primarily arise in vulnerable people inhabiting developing countries. ADP-ribosylation and the associated enzymes, including PARP and PARG, play an essential role in the ability of *Trypanosomatidae* to establish a successful infection in the human host. Given the lack of safe and effective medications available to combat these parasites, and the parasites’ reliance on the functioning ADP-ribosylation to maintain parasitic viability, the disruption of their ADP-ribosylation systems may present a novel and effective therapeutic option. However, a deep understanding of the molecular mechanisms underpinning the *Trypanosomatidae* ADP-ribosylation systems is required to accurately target it. As such, this review will offer insight into the current understanding of these systems and how they may be exploited to advance the options available for effective *Trypanosomatidae* treatment.

*T. cruzi* is the causative agent of Chagas disease, which is endemic to Central and South America. An initial infection during the acute stage results in flu-like systems. These symptoms mask the diagnosis of *T. cruzi* infection [38]. During chronic stages, an infection can lay dormant for decades, characterised by potentially minimal symptoms and low parasitemia, which causes issues in detecting the infection. The chronic infection eventually results in organ enlargement, with parasites primarily targeting the heart, leading to numerous cardiac complications and potential death. Chagas is a disease of poverty, meaning minimal financial gains in the development of tools can help to combat it, which is reflected in that only 0.67% of US funding for neglected diseases 2009–2019 was applied to Chagas [39]. These issues are compounded by the inefficacy of current treatments. Benznidazole and nifurtimox have limited cure rates and possess toxic side effects and the need for an effective alternative has been identified [40].

*T. brucei* is the etiological agent of African trypanosomiasis. Vectoral transmission is the most common route of infection, in which parasites enter the human host via bites inflicted by the primary tsetse fly host. Similar to Chagas disease, the initial stage of infection presents aspecific symptoms, followed by parasites eventually migrating to the brain, leading to neurological complications, and ultimately death, without proper treatment [41]. Available therapies for the neurological stage of infection are also limited, with Melarsoprol the only available drug, although this causes death in 5% of the patients who ingest it since Melarsoprol resistance is present in some strains [42]. The anti-parasitic Fexinidazole has shown activity against both the CNS and peripheral stages of African trypanosomiasis, although studies remain in clinical trials and effective drug alternatives are required to limit resistance [43].

Leishmaniasis is an umbrella term for three distinct diseases: visceral, cutaneous, and mucocutaneous leishmaniasis. Infections are predominantly found in Asia, the Middle East, and Northern Africa to differing degrees. The diseases are caused by several different *Leishmania* species and are transmitted commonly by bite wounds inflicted by an infected female phlebotomine sand fly. Visceral leishmaniasis is the most severe, as it causes a systemic infection that almost invariably is fatal without rapid treatment. The cutaneous infection leads to superficial skin lesions, whilst mucocutaneous infection leads to significant damage of the buccal and nasal cavities via the formation of damaging mucocutaneous lesions, which may disappear and reoccur repeatedly [44]. Given the severe nature of the infection, the majority of attention in the development of novel therapies for leishmaniasis has been focused on visceral leishmaniasis [45,46]. Currently, licenced therapies have several issues relating to their efficacy, dangerous side effects, and availability in endemic countries. Liposomal amphotericin B is often the first-choice drug, with miltefosine approved in 2014 by the FDA for oral treatment. Amphotericin B can cause significant side effects [47] and resistance against it has been documented, given that it is often a front-line drug for many fungal infections [48].

Consequently, given the scarcity of funding available to research the development of novel therapeutics to combat these NTDs and the lack of efficacy for existing treatments, the identification of novel ways to combat these infections is paramount. *T. cruzi*, *T. brucei*, and *Leishmania* all utilise ADP-ribosylation in several distinct ways to facilitate successful infection in the human host. Given the avenues with potential to be explored, whereby ADP-ribosylation manipulation can be harnessed to combat viral and bacterial infection, it is feasible that inhibiting or disrupting ADP-ribosylation in these parasites could lead to diminished parasite survival and proliferation.

In contrast to the extensive PARP network found in humans, the PARP system present in *Trypanosomatidae* is much simpler. *T. cruzi* and *T. brucei* both possess a singular PARP enzyme, designated TcPARP and TbPARP, respectively. Both parasites also utilise a sole poly(ADP-ribose) glycohydrolase (PARG) enzyme, which is used to reverse the action of PARP via the hydrolysis of ribose–ribose bonds present in PARP, which helps prevent extensive DNA damage by excessive PARP accumulation [49]. The primary ADP-ribosylation mechanisms within these parasites use polyADP-ribosylation, although there is also evidence of them using monoADP-ribosylation systems. Both *T. cruzi* and *T. brucei* possess proteins that are homologous to human MacroD1 and MacroD2, which are domains that hydrolyse and cleave ADP-ribose attachments to proteins in monoADP-ribosylation within human systems [50]. The homologous proteins present in *T. cruzi* and *T. brucei* (denoted TcMDO and TbMDO, respectively), both possess the core macrodomain fold present in human MacroD1/D2, although the N-terminus of the parasitic proteins differ heavily from the human MacroD1 and MacroD2, whereby they lack the adenine-binding region, indicating that TcMDO and TbMDO bind differently to the adenine moieties [51]. Given the nature of ADP-ribosylation in *Trypanosomes*, there is potential for the inhibition of TcMDO/TbMDO to have therapeutic penitential, yet there is a lack of inhibitors against macrodomain proteins [51]. 

## 3. ADP-Ribosylation in *Trypanosoma cruzi*

### 3.1. Use of ADP-Ribosylation and Associated PARP and PARG Enzymes in Trypanosoma cruzi

*Trypanosoma cruzi* expresses a sole PARP enzyme throughout its lifecycle, known as TcPARP (Figure 1). An initial study of TcPARP revealed several structural and molecular similarities to its human homologue, hPARP-1 [52]. Similar to hPARP-1, TcPARP is activated via DNA strand nicks, upon exposure of the parasite DNA to damaging agents such as H_2_O_2_. Initial studies on TcPARP revealed a highly evolutionarily conserved C-terminal catalytic domain that is homologous to *T. brucei* PARP, human PARP-1, and PARP-4. Furthermore, the catalytic triad of histidine, tyrosine, and glutamic acid is utilised for PAR elongation in hPARP-1, the closest human homologue to TcPARP is conserved within the parasite [53]. Though further structural similarities were identified between TcPARP and human PARPs (including the essential presence of glutamic acid to facilitate PARP activity) [54,55], TcPARP differentiates itself from human PARP through its method of DNA damage detection. Whilst DNA repair is essential for both humans and *Trypanosomatidae* to facilitate proper cellular function, human PARP enzymes utilise an N-terminal zinc-finger domain to bind DNA. This domain is seemingly absent in *T. cruzi*, which instead possesses an N-terminal region thought to be able to bind DNA through the large quantity of basic amino acid residues present in the domain (approximately 27% within the first 70 residues in TcPARP) [53], therefore, allowing *T. cruzi* to regulate DNA activity, which is corroborated by the ability of this region to bind hPARP-1 and hPARP-2 [56].

TcPARP activity is dependent upon the detection of DNA damage and is key in successful DNA repair and signalling. It serves roles in the ability of the parasite to differentiate into different life-cycle stages and establish infection in a human host. Consequently, the use of PARP inhibitors in the treatment of *T. cruzi* has been studied and has seen some success. Olaparib, a common PARP inhibitor, exerts strong inhibitory effects on hPARP-1 and TcPARP activities [52,57]. Olaparib has shown the ability to reduce amastigote (non-motile form lacking flagella) formation in a range of cell lines [52]. Another common hPARP inhibitor, EB-47 demonstrated an ability to prevent pADPr formation in human cell lines, although it failed to prevent pADPr formation in *T. cruzi* in vivo [58]. Villchez Larea and colleagues hypothesised that EB-47’s inability to adequately prevent pADPr formation in vivo could be due to EB-47’s comparatively large size and excessive polarity compared to other inhibitors. Significantly, Olaparib has been shown to reduce epimastigote growth in vitro by more than an 100× larger concentration of Nifurtimox (the established treatment modality for Chagas) [59]. Given the severity of the Nifurtimox side effects, the potential of PARP inhibitors as an alternative or combinatorial approach to Chagas therapy is plausible. However, specificity remains a challenge as TcPARP utilises the same nicotinamide binding site, to bind inhibitors, as the hPARP enzymes.

### 3.2. The Potential of ADP-Ribosylation Targeted Therapy for Trypanosoma cruzi

Intriguingly, there may be an interplay between hPARP-1 and TcPARP during infection establishment by *T. cruzi*. hPARP-1 silencing experiments in A549 cells demonstrated a significantly decreased amastigote number as well as a reduction in trypomastigotes in the culture media [52]. *T. cruzi* is dependent upon the ability of healthy trypomastigotes to establish a lasting infection and the subsequent differentiation and multiplication of amastigotes [60]. Therefore, successful inhibition and prevention of *T. cruzi* require the disruption of differentiation and growth. *T. cruzi* trypomastigote penetration of cardiac tissue in vitro leads to ROS generation, through activation of hPARP-1. ROS accumulation subsequently triggers the expression of NF-kB, which is a transcription factor that facilitates the transcription of multiple cytokines acting in concert to allow successful penetration of the parasite into cell lines in vitro [61,62]. The absence of cytokines TNF-α and IL-1β resulted in decreased levels of infection, demonstrating that successful infection by *T. cruzi* requires some activation of NF-kB [63]. Hence, this presents the interesting hypothesis that *T. cruzi* relies not only on TcPARP but also the activation of cytokine factors, mediated by hPARP-1. Further studies on the relationship between hPARP-1 mediated persistence of *T. cruzi* are required. This offers evidence for the importance of both innate and host PARP enzymes in *T. cruzi*, furthering the case of PARP inhibitors as a treatment for Chagas. It is also clear that hPARP-1 is likely the critical enzyme to target, given that hPARP-1 silencing results in higher inhibitory effects on infection than Olaparib treatment. This suggests successful infection relies upon a mechanism involving hPARP-1 specifically, as opposed to other hPARP enzyme members.

However, chronic exposure to PARP inhibitors can be profoundly detrimental. Should PARP inhibitors be assessed regarding Chagas therapy, protection from DNA damage is a fundamental consideration. PARG enzymes can reverse the effects of PARP enzymes, via clearance of excessive PAR accumulation to circumvent any harm to cells [64]. The PARG enzyme present in *T. cruzi*, denoted TcPARG, has been demonstrated as essential for epimastigote growth and the infection cycle in vitro (Figure 1). TcPARG shares 46.5% sequence identity with human PARG and possesses a preserved domain, including the tyrosine residues utilised to bind PARG inhibitors [65]. Importantly, as with hPARP-1, *T. cruzi* seemingly uses host PARG during infection establishment. A significantly lower number of intracellular amastigotes and infected cells were observed in hPARG knockout experiments compared to PARG inhibitor experiments [66]. The generation of pARPr via the exo-glycosidase activity of human PARG is thought to regulate factors associated with protein binding and post-translational modification [66]. This work revealed that *T. cruzi* may rely on both innate and host PARG factors to allow successful infection in the human host. Given PARG’s role in the removal of PARylation from cells, its inhibition would allow a longer and more impactful use of PARP inhibitors in Chagas treatment, but of course, the same considerations remain over limiting excess PARP-mediated DNA damage to the host [67]. As such, it seems apparent that there is a high level of interconnectivity between TcPARP, TcPARG, and hPARP/hPARG. Further investigation into this relationship may reveal the optimal avenues for PARP/PARG inhibition in treating Chagas infection, whilst preserving host cell homeostasis. However, *T. cruzi* relies on this relationship for infection establishment. The association between pathogen and human PARPs/PARG is open to intervention that may create new treatments to reduce infection and its associated clinical manifestations.

## 4. ADP-Ribosylation in *Trypanosoma brucei*

### 4.1. PARP Enzymes in Trypanosoma brucei and Potential Therapies

As in *T. cruzi*, *T. brucei* possesses one PARG and one PARP enzyme, denoted as TbPARG and TbPARP, respectively (Figure 1). TbPARP is highly conserved and similar to TcPARP and hPARP-1 in terms of its sequence, structure, and function [68,69]. A basic sequence of amino acids is likely responsible for DNA damage detection by TbPARP and the subsequent activity of the enzyme. Notably, TbPARP lacks a reliance on metal ions for TbPARP activity, both of which are characteristics that are shared with TcPARP. Metal ions (including Mn^2+^, Ni^2+^, and Zn^2+^) can have an inhibitory effect on the activity of both TbPARP and TcPARP, which is likely due to the metal ions binding to sulfhydryl groups, thought to be necessary for disulphide bond formation and reduction reactions [68]. This is in contrast to the impacts that metal ions have on human PARP enzymes, with Ca^2+^ and Mn^2+^ able to increase the levels of PAR synthesis, with human PARP enzymes relying on an alternative mechanism for oxygen reduction and disulphide bond formation [70]. Studies on TbPARP structure have also elucidated N-terminus disorder, which is a characteristic shared by both hPARP2 and hPARP3 [24]. TbPARP, along with TcPARP, has also been shown to have a WGR binding domain composed of arginine, glycine, and tryptophan, which is present in several eukaryotic organisms, and has a demonstrated role in DNA-dependent PARP activity for PARP enzymes lacking the characteristic zinc-finger binding domain of hPARP-1 [71]. As such, this offers further insight into the mechanisms underpinning the activity of TbPARP, along with the potential role of the WGR domain in TbPARP activation. The DNA-dependent activation of TbPARP specifically requires phosphorylated single-strand overhangs for the recognition of DNA nicks, with the WGR domain hypothesised to play a currently unknown role in this process [49].

Several studies have identified that the most efficacious inhibitors of TbPARP are the same as for TcPARP and hPARP-1 inhibition, most notably Olaparib, and EB-47 [71,72]. TbPARP shares the same nicotinamide-based NAD^+^ binding site as TcPARP and hPARP-1, explaining the consistency in inhibitor potency. There are differences between some inhibitors, such as Rucaparib, which is unfavourable for TbPARP binding, thought to be due to the presence of a serine residue in place of the alanine present in the binding site of TcPARP and hPARP-1 [71]. Whilst this sequence difference means that some inhibitors may be less useful in *T. brucei* treatment than for *T. cruzi*, the use of the existing inhibitors is still a realistic treatment option for African trypanosomiasis, given the broad activity of PARP inhibitors on TbPARP and the role of TbPARP in parasite development and proliferation. Despite the promise shown in vitro, reducing PAR synthesis in vivo in *T. brucei* with conventional inhibitors remains a challenge. As in *T. cruzi*, for an inhibitor to exert an effect on PARP activity and PAR formation in the parasite, the inhibitor must be both small and polar enough to successfully cross parasite membranes. 

TbPARP activation is dependent on the detection of DNA nicks and subsequent migration to the site of genomic damage in the nucleus, and TbPARP seemingly exerts similar protective effects to TcPARP in facilitating parasite growth and differentiation [73]. Moreover, excessive accumulation of PAR via TbPARP activity results in cellular damage and cell death [74]. 

### 4.2. PARG Enzymes in Trypanosoma brucei and Potential Therapies Targeting PARG Enzymes

The role of PARG-mediated suppression of PARP enzymes in *T. brucei* is unclear. However, a relationship between PARG and PARP enzymes exists within *T. brucei*. Therefore, although it is likely that a PARP/PARG-mediated treatment is closer for Chagas disease, a similar achievement could be attained for African trypanosomiasis. TbPARG shares 60% sequence similarity with human PARG, including the adenosine diphosphate hydroxymethyl pyrrolidinediol (ADP-HPD) binding site, with ADP-HPD commonly involved in PARG inhibition, meaning PARG inhibitors used in mammalian systems may offer relevance in *T. brucei* [65]. In *T. bruce*, the depletion of PARG has been shown to result in the increased nuclear accumulation of PAR in *Trypanosomes*, even in the absence of oxidative stress [75]. The duality of PARP and PARG enzymes likely exerts a similar effect in *T. brucei* as in *T. cruzi*, and their regulation strikes a fine balance between cellular protection and cell death. PARP enzymes exert protective effects against DNA damage in these parasites, but excessive PAR accumulation can lead to disrupted DNA repair, and NAD^+^ and ATP depletion resulting in apoptosis. PARP and PARG enzymes remain credible targets for intervention in *T. brucei* as the parasite relies on both enzymes to establish infection. Nevertheless, the current state of understanding is poor in *T. brucei* in comparison to *T. cruzi*. Further work is required to elucidate the specific intricacies of the mechanisms involved in *T. brucei*, with therapies potentially requiring slight adjustments given the difference in the sequence of the nicotinamide binding site, of TbPARP specifically, compared to TcPARP and hPARP-1.

It has been hypothesised that TbPARP is less important to *T. brucei* parasite viability than TcPARP is to *T. cruzi* [68]. Nonhomologous end joining (NHEJ) repair of DNA breaks is not present in *Trypanosomatidae*, as it is in eukaryotes, and as such, double-strand repairs within these parasites rely on microhomology-mediated end joining (MMEJ) [76]. This DNA repair mechanism relies upon small homologous regions within the broken ends to align and repair the strands. This more commonly results in sequence deletions and other modifications than in NHEJ. MMEJ is often viewed as a less preferred alternative to NHEJ because of this, yet MMEJ is omnipresent in DNA repair for *T. brucei*, although is it not understood. Unlike the roles of TcPARP and hPARP-1, the role of TbPARP is likely more similar to hPARP2 and hPARP3, whereby it is used in specific DNA repair pathways and not as a universal DNA repair enzyme, evidenced by its specific role within MMEJ. Given the lack of necessity of TbPARP for *T. brucei* viability, further study is warranted to fully understand the role of TbPARP in DNA repair; however, currently, it would seem that attention may be better focused on exploiting PARG enzymes to treat African trypanosomiasis, given the strong effect PARG depletion has on decreasing parasite viability.

## 5. ADP-Ribosylation in *Leishmania*

### 5.1. LdARL-3A Ribosylation Factor as a Therapeutic Target in Leishmania

ADP-ribosylation in *Leishmania* is less studied and poorly understood compared to the *Trypanosomes*, though work has taken place to better understand the importance of ADP-ribosylation in relation to *Leishmania* viability and infection. LdARL-3A is an ADP-ribosylation factor identified in *Leishmania donovani*, which is expressed specifically during the promastigote stage, during the insect stage of the lifecycle. ARL (ADP ribosylation-like) enzymes are essential for numerous cellular processes, including trafficking, endocytosis, and other cell signalling pathways. LdARL-3A seemingly plays a role in flagellum formation and viability, as overexpression of a constitutively active LdARL-3A mutant led to a correlated decrease in flagellum length, with stronger levels of overexpression leading to larger decreases in flagellar length [77]. Therefore, if the function of LdARL-3A can be inhibited, the motility of the parasite in the insect host may be reduced before the infection of the human host. Further study of LdARL-3A revealed that LdARL-3A possesses two forms, a GDP-bound and a GTP-bound form [78]. The GDP-bound form is considered inactive and is the form in which overexpression led to parasite death in the stationary phase, whereas the GTP-bound form is active and overexpression led to diminished flagellar length. The disruption of LdARL-3A, with it switching between the inactive and active forms, is what leads to the reduction in flagellar length. Hence, this evidence suggests that LdARL-3A is a potential drug target, in the attempt to prevent transmission from the insect vector to the human host, by inhibiting the motility by decreasing the flagellar length.

### 5.2. LiSIR2RP1 as a Therapeutic Target in Leishmania

*Leishmania infantum* possesses a gene denoted LiSIR2RP1 that encodes the SIR2 protein, which is a deacetylase to several cellular substrates, including histone lysine residues [79]. This deacetylation activity is dependent upon the use of NAD^+^. Disruption of LiSIR2RP1 has been shown to decrease the viability of *Leishmania infantum* amastigotes both in vivo and in vitro, with the close association of LiSIR2RP1 with the cytoskeletal structure of both *L. infantum* promastigote and amastigotes. LiSIR2RP1 can exert deacetylase activity upon tubulin, which is critical for parasite structural viability as well as the parasite’s ability to interact with host cells [80]. *Leishmania* tubules and microtubules play an essential part in successful parasite division and structural integrity; thus, if LiSIR2RP1 can be further explored, it could yield a better understanding of the role LiSIR2RP1 plays in parasite integrity and remodelling and offer potential therapeutic alternatives via LiSIR2RP1 targeting.

### 5.3. Targeting NAD+ Salvage Pathways as a Therapeutic Option in Leishmania

*Leishmania* species are NAD^+^ auxotrophs and require NAD^+^ sequestered from host cells. *Leishmania* lacks both intrinsic de novo pathways that can be used for innate biosynthesis of NAD, which use either L-tryptophan or aspartic acid as a precursor (which are in eukaryotic and prokaryotic de novo pathways, respectively). *Leishmania* instead uses NR (nicotinamide riboside) as a precursor for NAD^+^ production in salvage pathways, wherein NAD^+^ is sequestered from the host [81]. The NAD nucelotidase (NadN) enzyme first described in *Haemophilus influenzae* was found to also be present in *Leishmania* [82]. NadN is a periplasmic enzyme thought to be involved in NAD^+^ synthesis via NAD^+^ hydrolysis into NR, adenosine, and phosphate, and finally, NR uptake across the inner membrane of the parasite and NadR catalysed resynthesis of NR into NAD^+^ [83]. Given the high level of conformation change identified in the enzymatic pathway, it has been identified as a possible target for inhibition to prevent parasite replication, which is corroborated by NadN knockout experiments in *Leishmania* resulting in a significant NAD^+^ concentration decrease, which coincided with reduced parasite proliferation and virulence.

## 6. Summary

It is apparent that ADP-ribosylation plays a dynamic and important role in infection establishment for *Trypanosomatidae*, and there exists a considerable interplay between these parasites’ innate ADP-ribosylation systems and the role of the host ADP-ribosylation systems, which exerts protective effects within the host to ward off infection. Given that *Trypanosomatidae* relies on ADP-ribosylation to maintain parasite viability, exploitation of these systems within the parasites offers an attractive avenue to explore novel therapies to prevent or treat the infection. This is especially important given the minimal current treatment options for all *Trypanosomatidae*.

*T. cruzi*, in particular, is reliant on the proper function of its innate PARP and PARG enzymes to be able to maintain an infection within a human host. Further evidence is required to be able to fully understand the interactions between humans and *T. cruzi* PARP and PARG enzymes. There exists the possibility of inhibiting both TcPARP and TcPARG to dysregulate *T. cruzi* function, as evidenced by the lack of parasite viability in TcPARP and TcPARG knockout experiments. Interestingly for *T. cruzi*, the knockout of hPARP-1 and hPARG also yielded lower parasite viability, in terms of lower numbers of amastigotes and trypomastigotes in culture, revealing that *T. cruzi* is also reliant on the action of hPARP-1 and hPARG, to some extent, highlighting the multifaceted way in which *T. cruzi* infection can potentially be disrupted. It is necessary to fully understand the mechanisms by which PARP and PARG may be used therapeutically, as the right balance needs to be struck for the host to benefit from this therapy without harm. Inhibiting PARP can disrupt infection, but PARP inhibition leads to detrimental impacts for the host; therefore, being able to clear the accumulation of PAR with PARG enzymes would be required. However, PARG enzymes could also be a target for inhibition themselves, given that TcPARG and hPARG are required for infection. TcPARP, TcPARG, hPARP, and hPARG are all credible targets for inhibition given their important roles in Chagas infection establishment, yet their close relationship necessitates fine-tuning of any potential PARP/PARG therapy against *T. cruzi*.

*T. brucei* similarly possesses innate PARP and PARG enzymes (TbPARP and TbPARG). However, unlike in *T. cruzi*, TbPARP is seemingly less essential for successful infection, though TbPARG maintains an essential role and is, therefore, likely the better candidate for targeted therapy to prevent successful *T. brucei* infection. Overall, the understanding of the interplay between the ADP-ribosylation factors of *T. brucei* is likely not as advanced as in *T. cruzi*. This highlights the need for subsequent research to better understand how ADP-ribosylation in *T. brucei* could be exploited therapeutically. TbPARG inhibition decreases *T. brucei* viability significantly through the reduced numbers of parasites, which makes TbPARG the leading candidate for therapy. TbPARP plays a less focal role in DNA repair than hPARP-1 and TcPARP and is, therefore, likely to not be the best candidate for therapy, although it may be used in a combinatorial approach when targeting TbPARG. Given the close genetics of *T. cruzi* and *T. brucei* and the similarities in their ADP-ribosylation systems, it may be worth exploring how or if *T. brucei* may compromise hPARP/hPARG enzymes to establish infection in a human host.

Given the status of *Leishmania* as an NAD^+^ auxotroph, a therapeutic option for this genus is likely to be inhibiting their ability to harvest NAD^+^ from the host via disruption of their NAD^+^ salvage pathways. *Leishmania* uses NMN and NR as essential NAD^+^ precursors, with the essential enzyme NadN being active in the pathway. Inhibition of NadN is another likely therapeutic target, given its specificity to *Leishmania* and other pathogens, and its critical role in allowing *Leishmania* to access NAD^+^, necessary for several essential cellular processes that maintain parasite viability. Alternatively, LdARL3A and LiSIR2RP1 are essential genes for flagellar growth and parasite cytoskeletal maintenance, respectively. Knockout experiments of LdARL3A decreased flagellar growth to the extent that parasite movement could be prevented, which would aid in infection prevention as the parasite is unable to disseminate in the host. Knockout LiSIR2RP1 experiments decreased the number of viable parasites, likely due to the close relationship of LiSIR2RP1 to the cytoskeletal structure of *Leishmania infantum*. However, LdARL3A and LiSIR2RP1 are both specific to one species of *Leishmania* (*L. donovani* and *L. infantum*, respectively) and consequently, these approaches may not be universal for all *Leishmania* species, and so further research is needed to fully understand the conservancy of these genes across medically relevant *Leishmania* species, and if targeting these genes may require differing species-level approaches. Therefore, the universality of targeting NAD^+^ salvaging for all *Leishmania* species likely presents the best current option for an ADP-ribosylation-related therapy in these parasites.

## Figures and Tables

**Figure 1 pathogens-12-00708-f001:**
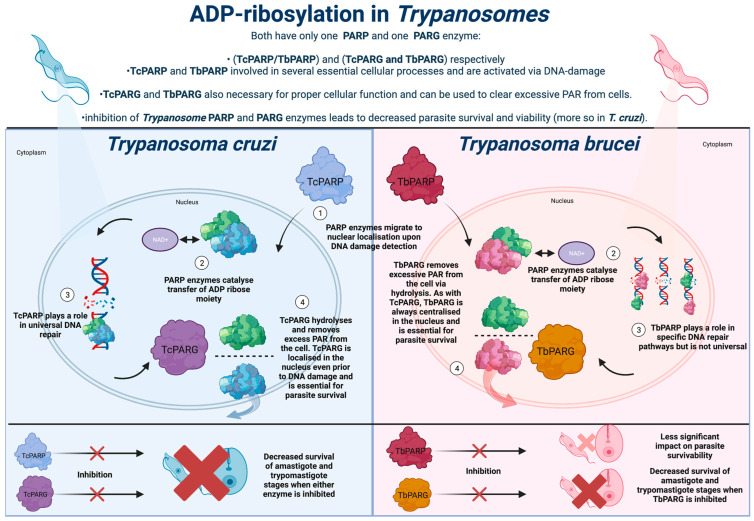
**ADP-ribosylation in Trypanosomes.** In contrast to pathways in eukaryotes, ADP-ribosylation in *Trypanosomatidae* is more linear, with the parasites only expressing one PARP enzyme. If this pathway can be understood and exploited, it may reveal new avenues for combatting *Trypanosomatidae* infection.

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
