# Peer review of "Roles of ADP-Ribosylation during Infection Establishment by Trypanosomatidae Parasites"

_pathogens, 2023, doi:10.3390/pathogens12050708_

Round 1
Reviewer 1 Report
In this manuscript submitted by Dowling et al, summarizes the role of ADP-ribosylation during infection by Trypanosomatidae parasites. The authors discuss the potential therapeutic interventions targeting ADP-ribosylation pathway among these parasites. The authors started with the role of ADP-ribosylation in infection on a broader note in viral infection, then discusses Tryapanosoma and Leishmania. Briefly concluded the review pointing towards the use of ADP-ribosylation inhibitors and combinational treatment with existing therapeutic options.
While this manuscript is written with thoughtful points at some places, at most of the places the manuscript needs to be improved to articulate the points. Particularly, some facts about ADP-ribosylation are not clear.
Some of the major comments:
1. In the abstract, the authors mentioned that “currently licensed medications are outdated”. It would be easier to list out those in a table.
2. At many places, the authors stated that compared to human PARPs, however; PARPs and PARG among human vs bacteria vs trypanasoma was not discussed. For example, an introduction paragraph discussing ADP-ribosylation should be included. Also, PARG is one of the enzymes to remove PAR chain, but there are some PAR hydrolases present in human cells. It was not mentioned. Poly vs Mono ADP-ribosylation also not clearly described, particularly in trypanasoma. While on line 40, the authors mentioned that PARP form branching polymers, some PARPs do form only linear polymers.
3. The heading “1. ADP-ribosylation in infection” summarizes PARP role in virus and bacteria. It would be nicer to add subheading and discuss these
4. The heading “2. Trypanosomatidae” introduces the parasite, occurrence, and etiology. The authors have not mentioned what is the closest human PARP for TcPARP or TbPARP and the sequence similarity, domain similarity, and a graphical scheme would be helpful.
5. A subheading under each parasite and clearly distinguishes how these parasites utilizes ADP-ribosylation for infection, the potential drug treatment would be easier to follow.
The writing should be improved. At many parts, they are bit and pieces.
Reviewer 2 Report
The manuscript aims to summarize current understanding on ADP-ribosylation in Trypanosomatidae during human infection, and highlight the impacts of ADP-ribosylation on infection establishment. The concept somehow is very interesting, however the review was not well organized. Essential references supporting author's points are missing, the importance and mechanism of ADP-ribosylation upon establishment of parasites infection are not well documented and lacking details.
lots of misspelling need to be corrected, like H202, NF-kb.
Reviewer 3 Report
The well-structured review clearly outlined the roles of ADP-ribosylation in the establishment of Typanosomatosis parasite infection. In the review, authors detailed the roles of poly(ADP-ribose) polymerase (PARP)-mediated ADP ribosylation in viral and bacterial infection of host cells, and discussed the ADP-ribosylation in Trypanosomatidae as well as its association with the viability of the parasites. This review will make valuable contributions to the efficient treatment of Trypanosomatidae. I recommend it for publication after considering the following suggestions/comments.
Issues:
1. Introduction part missing: Authors should include an introduction part in the main text to display the overview of the topic, here the roles of ADP ribosylation during Trypanosomatidae parasite infection, and to explain why a review of the topic is necessary.
2. ADP-Ribosylation in infection: Authors provided a comprehensive overview of the roles of host PARP enzymes in viral or bacterial infection. Are there any studies showing a role of host PARG protein in viral or bacterial infection? Even if not, authors should also mention PARG in this section to give a more comprehensive introduction of poly(ADP-ribosyl)ation (PARylation) and its modulators.
3. ADP-Ribosylation in Trypanosoma cruzi: hPARG knockout experiment gave a significant lower number of intracellular amastigotes and infected cells compared to PARG inhibitor experiment. How about their influence on the viability of host cells? Is there any explanation for the mechanism of this phenomenon? Dose it mean that excess ADP ribosylation can significantly influence the infection of the parasites?
4. ADP-Ribosylation in Trypanosoma cruzi: Authors stated that “Given PARG’s role in removal of PARP from cells, inhibition of PARG would allow a longer and more impactful use of PARP inhibitors in Chagas treatment, but of course the same considerations remain over limiting excess PARP mediated DNA damage to the host.” Due to the specificity of PARP inhibitors, inhibition of PARG may have great impacts on the cellular pathways of other PARP proteins. This is also an aspect worth considering.
1. Abstract part, line 15: Please remove “;”.
2. Page 4, line 186: Please change “H202” to “H2O2” with “2” subscripted.
3. Page 5, line 249: Please change “Given PARG’s role in removal of PARP from cells, …” to “Given PARG’s role in removal of PARylation from cells, …”
Reviewer 4 Report
Current review mainly focus on the importance of ADP-ribosylation in Trypanosomatidae during human infection, and how ADP-ribosylation regulates it's infectivity. Most of the part of the review is nicely represent the regulatory role of ADP-ribosylation in human pathogenesis. Major concern for the current form of the review is as follows:
1) Author should recheck the formatting of text ( spacings, brackets). It's very poorly formatted.
2) Needs to add references at Line 41, 56 and 179.
3) Replace "in with In" at line 200.
4) Rephrase line 30-31.
It's well written in most of the part.
Round 2
Reviewer 1 Report
The authors addressed all the comments raised.
Very minor English correction is required.
Author Response
Thank you for your comments.
Reviewer 2 Report
The authors have addressed all my concerns, and the manuscript should be able to be published.
Author Response
Thank you for your comments